# Global MAP-Optimality by Shrinking the Combinatorial Search Area with Convex Relaxation

**Bogdan Savchynskyy**[1]    **Jörg Kappes**[2]    **Paul Swoboda**[2]    **Christoph Schnörr**[1,2]

[1]Heidelberg Collaboratory for Image Processing, Heidelberg University, Germany
`bogdan.savchynskyy@iwr.uni-heidelberg.de`
[2]Image and Pattern Analysis Group, Heidelberg University, Germany
`{kappes,swoboda,schnoerr}@math.uni-heidelberg.de`

## Abstract

We consider energy minimization for undirected graphical models, also known as the MAP-inference problem for Markov random fields. Although combinatorial methods, which return a provably optimal integral solution of the problem, made a significant progress in the past decade, they are still typically unable to cope with large-scale datasets. On the other hand, large scale datasets are often defined on sparse graphs and convex relaxation methods, such as linear programming relaxations then provide good approximations to integral solutions.

We propose a novel method of combining combinatorial and convex programming techniques to obtain a global solution of the initial combinatorial problem. Based on the information obtained from the solution of the convex relaxation, our method confines application of the combinatorial solver to a small fraction of the initial graphical model, which allows to optimally solve much larger problems. We demonstrate the efficacy of our approach on a computer vision energy minimization benchmark.

## 1 Introduction

The focus of this paper is energy minimization for Markov random fields. In the most common pairwise case this problem reads

$$\min_{x \in \mathcal{X}_\mathcal{G}} E_{\mathcal{G},\theta}(x) := \min_{x \in \mathcal{X}_\mathcal{G}} \sum_{v \in \mathcal{V}_\mathcal{G}} \theta_v(x_v) + \sum_{uv \in \mathcal{E}_\mathcal{G}} \theta_{uv}(x_u, x_v), \tag{1}$$

where $\mathcal{G} = (\mathcal{V}_\mathcal{G}, \mathcal{E}_\mathcal{G})$ denotes an undirected graph with the set of nodes $\mathcal{V}_\mathcal{G} \ni v$ and the set of edges $\mathcal{E}_\mathcal{G} \ni uv$; variables $x_v$ belong to the finite *label* sets $\mathcal{X}_v$, $v \in \mathcal{V}_\mathcal{G}$; *potentials* $\theta_v \colon \mathcal{X}_v \to \mathbb{R}$, $\theta_{uv} \colon \mathcal{X}_u \times \mathcal{X}_v \to \mathbb{R}$, $v \in \mathcal{V}_\mathcal{G}$, $uv \in \mathcal{E}_\mathcal{G}$, are associated with the nodes and the edges of $\mathcal{G}$ respectively. We denote by $\mathcal{X}_\mathcal{G}$ the Cartesian product $\otimes_{v \in \mathcal{V}_\mathcal{G}} \mathcal{X}_v$.

Problem (1) is known to be NP-hard in general, hence existing methods either consider its convex relaxations or/and apply combinatorial techniques such as branch-and-bound, combinatorial search, cutting plane etc. on top of convex relaxations. The main contribution of this paper is *a novel* method to combine convex and combinatorial approaches to compute a *provably* optimal solution. The method is very general in the sense that it is not restricted to a specific convex programming or combinatorial algorithm, although some algorithms are more preferable than others. The main restriction of the method is the neighborhood structure of the graph $\mathcal{G}$: it has to be sparse. Basic grid graphs of image data provide examples satisfying this requirement. The method is applicable also to higher-order problems, defined on so called factor graphs [1], however we will concentrate mainly on the pairwise case to keep our exposition simple.

**Underlying idea.** Fig. 1 demonstrates the main idea of our method. Let $\mathcal{A}$ and $\mathcal{B}$ be two subgraphs covering $\mathcal{G}$. Select them so that the only common nodes of these subgraphs lie on their mutual border

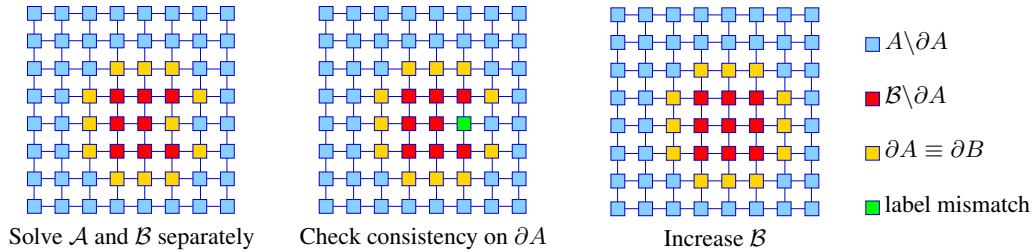

| | |
|---|---|
| | $A \backslash \partial A$ |
| | $\mathcal{B} \backslash \partial A$ |
| | $\partial A \equiv \partial B$ |
| | label mismatch |

Solve $\mathcal{A}$ and $\mathcal{B}$ separately    Check consistency on $\partial A$    Increase $\mathcal{B}$

Figure 1: Underlying idea of the proposed method: the initial graph is split into two subgraphs $\mathcal{A}$ (blue+yellow) and $\mathcal{B}$ (red+yellow), assigned to a convex and a combinatorial solver respectively. If the integral solutions provided by both solvers do not coincide on the common border $\partial A$ (yellow) of the two subgraphs, the subgraph $\mathcal{B}$ is increased by appending mismatching nodes (green) and the border is adjusted respectively.

$\partial A (\equiv \partial B)$ defined in terms of the master-graph $\mathcal{G}$. Let $x_\mathcal{A}^*$ and $x_\mathcal{B}^*$ be optimal labelings computed independently on $\mathcal{A}$ and $\mathcal{B}$. If these labelings coincide on the border $\partial A$, then under some additional conditions the concatenation of $x_\mathcal{A}^*$ and $x_\mathcal{B}^*$ is an optimal labeling for the initial problem (1), as we show in Section 3 (see Theorem 1).

We select the subgraph $\mathcal{A}$ such that it contains a "simple" part of the problem, for which the convex relaxation is tight. This part is assigned to the respective convex program solver. The subgraph $\mathcal{B}$ contains in contrast the difficult, combinatorial subproblem and is assigned to a combinatorial solver. If the labelings $x_\mathcal{A}^*$ and $x_\mathcal{B}^*$ do not coincide on some border node $v \in \partial A$, we (i) increase the subgraph $\mathcal{B}$ by appending the node $v$ and edges from $v$ to $\mathcal{B}$, (ii) correspondingly decrease $\mathcal{A}$ and (iii) recompute $x_\mathcal{A}^*$ and $x_\mathcal{B}^*$. This process is repeated until either labelings $x_\mathcal{A}^*$ and $x_\mathcal{B}^*$ coincide on the border or $\mathcal{B}$ equals $\mathcal{G}$. The sparsity of $\mathcal{G}$ is required to avoid fast growth of the subgraph $\mathcal{B}$.

We refer to Section 3 for a detailed description of the algorithm, where we in particular specify the initial selection of the subgraphs $\mathcal{A}$ and $\mathcal{B}$ and the methods for (i) encouraging consistency of $x_\mathcal{A}^*$ and $x_\mathcal{B}^*$ on the boundary $\partial A$ and (ii) providing equivalent results with just a single run of the convex relaxation solver. These techniques will be described for the *local polytope* relaxation, known also as *a linear programming relaxation* of (1) [2, 3].

**Related work.** The literature on problem (1) is very broad, both regarding convex programming and combinatorial methods. Here we will concentrate on the local polytope relaxation, that is essential to our approach.

The local polytope relaxation (LP) of (1) was proposed and analyzed in [4] (see also the recent review [2]). An alternative view on the same relaxation was proposed in [5]. This view appeared to be very close to the idea of the Lagrangian or dual decomposition technique (see [6] for applications to (1)). This idea stimulated development of efficient solvers for convex relaxations of (1). Scalable solvers for the LP relaxation became a hot topic in recent years [7–14]. The algorithms however, which guarantee attainment of the optimum of the convex relaxation at least theoretically, are quite slow in practice, see e.g. comparisons in [11, 15]. Remarkably, the fastest scalable algorithms for convex relaxations are based on coordinate descent: the diffusion algorithm [2] known from the seventies and especially its dual decomposition based variant TRW-S [16]. There are other closely related methods [17, 18] based on the same principle. Although these algorithms do not guarantee attainment of the optimum, they converge [19] to points fulfilling a condition known as *arc consistency* [2] or *weak tree agreement* [16]. We show in Section 3 that this condition plays a significant role for our approach. It is a common observation that in the case of sparse graphs and/or strong evidence of the unary terms $\theta_v$, $v \in \mathcal{V}_\mathcal{G}$, the approximate solutions delivered by such solvers are quite good from the practical viewpoint. The belief, that these solutions are close to optimal ones is evidenced by numerical bounds, which these solvers provide as a byproduct.

The techniques used in *combinatorial solvers* specialized to problem (1) include most of the classical tools: cutting plane, combinatorial search and branch-and-bound methods were adapted to the problem (1). The ideas of the cutting plane method form the basis for tightening the LP relaxation within the dual decomposition framework (see the recent review [20] and references therein) and for finding an exact solution for Potts models [21], which is a special class of problem (1). Combinatorial search methods with dynamic programming based heuristics were successfully applied

to problems defined on dense and fully connected but small graphs [22]. The specialized branch-and-bound solvers [23, 24] also use convex (mostly LP) relaxations and/or a dynamic programming technique to produce bounds in the course of the combinatorial search [25]. However the reported applicability of most combinatorial solvers nowadays is limited to small graphs. Specialized solvers like [21] scale much better, but are focused on a certain narrow class of problems.

The goal of this work is to employ the fact, that local polytope solvers provide good approximate solutions and to restrict computational efforts of combinatorial solvers to a relatively small, and hence tractable part of the initial problem.

**Contribution.** We propose a novel method for obtaining a globally optimal solution of the energy minimization problem (1) for sparse graphs and demonstrate its performance on a series of large-scale benchmark datasets. We were able to

- solve previously unsolved large-scale problems of several different types, and
- attain optimal solutions of hard instances of Potts models *an order of magnitude* faster than specialized state of the art algorithms [21].

For an evaluation of our method we use datasets from the very recent benchmark [15].

**Paper structure.** In Section 2 we provide the definitions for the local polytope relaxation and arc consistency. Section 3 is devoted to the specification of our algorithm. In Sections 4 and 5 we provide results of the experimental evaluation and conclusions.

## 2 Preliminaries

**Notation.** A vector $x$ with coordinates $x_v$, $v \in \mathcal{V}_\mathcal{G}$, will be called *labeling* and its coordinates $x_v \in \mathcal{X}_v$ – *labels*. The notation $x|_\mathcal{W}, \mathcal{W} \subset \mathcal{V}_\mathcal{G}$ stands for the restriction of $x$ to the subset $\mathcal{W}$, i.e. for the subvector $(x_v, \ v \in \mathcal{W})$. To shorten notation we will sometimes write $x_{uv} \in \mathcal{X}_{uv}$ in place of $(x_v, x_u) \in \mathcal{X}_u \times \mathcal{X}_v$ for $(v, u) \in \mathcal{E}_\mathcal{G}$. Let also $\mathbf{nb}(v)$, $v \in \mathcal{V}_\mathcal{G}$, denote the set of neighbors of node $v$, that is the set $\{u \in \mathcal{V}_\mathcal{G} : uv \in \mathcal{E}_\mathcal{G}\}$.

**LP relaxation.** The *local polytope* relaxation of (1) reads (see e.g. [2])

$$\min_{\mu \geq 0} \sum_{v \in \mathcal{V}_\mathcal{G}} \sum_{x_v \in \mathcal{X}_v} \theta_v(x_v)\mu_v(x_v) + \sum_{uv \in \mathcal{E}_\mathcal{G}} \sum_{(x_u, x_v) \in \mathcal{X}_{uv}} \theta_{uv}(x_u, x_v)\mu_{uv}(x_u, x_v)$$

$$\text{s.t.} \quad \begin{array}{l} \sum_{x_v \in \mathcal{V}_\mathcal{G}} \mu_v(x_v) = 1, \ v \in \mathcal{V}_\mathcal{G} \\ \sum_{x_v \in \mathcal{V}_\mathcal{G}} \mu_{uv}(x_u, x_v) = \mu_u(x_u), \ x_u \in \mathcal{X}_u, \ uv \in \mathcal{E}_\mathcal{G} \\ \sum_{x_u \in \mathcal{V}_\mathcal{G}} \mu_{uv}(x_u, x_v) = \mu_v(x_v), \ x_v \in \mathcal{X}_v, \ uv \in \mathcal{E}_\mathcal{G} \,. \end{array} \tag{2}$$

This formulation is based on the overcomplete representation of indicator vectors $\mu$ constrained to the *local polytope* commonly used for discrete graphical models [3]. It is well-known that the local polytope constitutes an outer bound (relaxation) of the convex hull of all indicator vectors of labelings (marginal polytope; cf. [3]).

The Lagrange dual of (2) reads

$$\max_{\phi, \gamma} \sum_{v \in \mathcal{V}_\mathcal{G}} \gamma_v + \sum_{uv \in \mathcal{E}_\mathcal{G}} \gamma_{uv} \tag{3}$$

$$\text{s.t.} \quad \begin{array}{ll} \gamma_v \leq & \tilde{\theta}_v^\phi(x_v) := \quad \theta_v(x_v) - \sum_{u \in \mathbf{nb}(v)} \phi_{v,u}(x_v), & v \in \mathcal{V}_\mathcal{G}, \ x_v \in \mathcal{X}_v \,, \\ \gamma_{uv} \leq & \tilde{\theta}_{uv}^\phi(x_u, x_v) := \quad \theta_{uv}(x_u, x_v) + \phi_{v,u}(x_v) + \phi_{u,v}(x_u), & uv \in \mathcal{E}_\mathcal{G}, \ (x_u, x_v) \in \mathcal{X}_{uv} \,. \end{array}$$

In the constraints of (3) we introduced the reparametrized potentials $\tilde{\theta}^\phi$. One can see, that for any values of the dual variables $\phi$ *the reparametrized energy* $E_{\tilde{\theta}^\phi, \mathcal{G}}(x)$ is equal to the non-parametrized one $E_{\theta, \mathcal{G}}(x)$ for any labeling $x \in \mathcal{X}_\mathcal{G}$. The objective function of the dual problem is equal to $D(\phi) := \sum_{v \in \mathcal{V}_\mathcal{G}} \tilde{\theta}_v^\phi(x_v') + \sum_{uv \in \mathcal{E}_\mathcal{G}} \tilde{\theta}_{uv}^\phi(x_{uv}')$, where $x_w' \in \arg\min_{x_w \in \mathcal{X}_v \cup \mathcal{X}_{uv}} \tilde{\theta}_w^\phi(x_w)$. A *reparametrization*, that is reparametrized potentials $\tilde{\theta}^\phi$, will be called *optimal*, if the corresponding $\phi$ is the solution of the dual problem (3). In general neither the optimal $\phi$ is unique nor the optimal reparametrization.

**Definition 1** (Strict arc consistency). *We will call the node $v \in \mathcal{V}_\mathcal{G}$ strictly arc consistent w.r.t. potentials $\theta$ if there exist labels $x'_v \in \mathcal{X}_v$ and $x'_u \in \mathcal{X}_u$ for all $u \in \text{nb}(v)$, such that $\theta_v(x'_v) < \theta_v(x_v)$ for all $x_v \in \mathcal{X}_v \backslash \{x'_v\}$ and $\theta_{vu}(x'_v, x'_u) < \theta_{vu}(x_v, x_u)$ for all $(x_v, x_u) \in \mathcal{X}_{vu} \backslash \{(x'_v, x'_u)\}$. The label $x'_v$ will be called* locally optimal.

If all nodes $v \in \mathcal{V}_\mathcal{G}$ are strictly arc consistent w.r.t. the potentials $\tilde{\theta}^\phi$, the dual objective value $D(\phi)$ becomes equal to the energy

$$D(\phi) = E_{\mathcal{G}, \tilde{\theta}^\phi}(x') = E_{\mathcal{G}, \theta}(x') \tag{4}$$

of the labeling $x'$ constructed by the corresponding locally optimal labels. From duality it follows, that $D(\phi)$ is a lower bound for energies of all labelings $E_{\mathcal{G}, \theta}(x)$, $x \in \mathcal{X}_\mathcal{G}$. Hence attainment of equality (4) shows that (i) $\phi$ is the solution of the dual problem (3) and (ii) $x'$ is the solution of both the energy minimization problem (1) and its relaxation (2).

Strict arc consistency of all nodes is sufficient, but not necessary for attaining the optimum of the dual objective (3). Its fulfillment means that our LP relaxation is tight, which is not always the case. However, in many practical cases the optimal reparametrization $\phi$ corresponds to strict arc consistency of *a significant portion of, but not all* graph nodes. The remaining non-consistent part is often much smaller and consists of many separate "islands". The strict arc consistency of a certain node $v$, even for the optimally reparametrized potentials $\tilde{\theta}^\phi$, does *not* guarantee global optimality of the corresponding locally optimal label $x_v$ (unless it holds for all nodes), though it is a good and widely used heuristic to obtain an approximate solution of the non-relaxed problem (1). In this work we provide an algorithm, which is able to prove this optimality or discard it. The algorithm applies combinatorial optimization techniques only to the arc inconsistent part of the model, which is often much smaller than the whole model in applications.

**Remark 1.** *Efficient dual decomposition based algorithms optimize dual functions, which differ from (4) (see e.g. [6, 13, 16]), but are equivalent to it in the sense of equal optimal values. Getting reparametrizations $\tilde{\theta}^\phi$ is less straightforward in these cases, but can be efficiently computed (see e.g. [16, Sec. 2.2]).*

## 3 Algorithm description

The graph $\mathcal{A} = (\mathcal{V}_\mathcal{A}, \mathcal{E}_\mathcal{A})$ will be called *an (induced) subgraph* of the graph $\mathcal{G} = (\mathcal{V}_\mathcal{G}, \mathcal{E}_\mathcal{G})$, if $\mathcal{V}_\mathcal{A} \subset \mathcal{V}_\mathcal{G}$ and $\mathcal{E}_\mathcal{A} = \{uv \in \mathcal{E}_\mathcal{G} : u, v \in \mathcal{V}_\mathcal{A}\}$. The graph $\mathcal{G}$ will be called *supergraph* of $\mathcal{A}$. The subgraph $\partial A$ induced by a set of nodes $\mathcal{V}_{\partial A}$ of the graph $\mathcal{A}$, which are connected to $\mathcal{V}_\mathcal{G} \backslash \mathcal{V}_\mathcal{A}$, is called its *boundary* w.r.t. $\mathcal{G}$, i.e. $\mathcal{V}_{\partial A} = \{v \in \mathcal{V}_\mathcal{A} : \exists uv \in \mathcal{E}_\mathcal{G} : u \in \mathcal{V}_\mathcal{G} \backslash \mathcal{V}_\mathcal{A}\}$. The complement $\mathcal{B}$ to $\mathcal{A} \backslash \partial A$, given by $\mathcal{V}_\mathcal{B} = \{v \in \mathcal{V}_\mathcal{G} : v \in \partial A \cup (\mathcal{V}_\mathcal{G} \backslash \mathcal{V}_\mathcal{A})\}$, $\mathcal{E}_\mathcal{B} = \{uv \in \mathcal{E}_\mathcal{G} : u, v \in \mathcal{V}_\mathcal{B}\}$, is called *boundary complement* to $\mathcal{A}$ w.r.t. the graph $\mathcal{G}$. Let $\mathcal{A}$ be a subgraph of $\mathcal{G}$ and potentials $\theta_v$, $v \in \mathcal{V}_\mathcal{G}$, and $\theta_{uv} \in \mathcal{E}_\mathcal{G}$ be associated with nodes and edges of $\mathcal{G}$ respectively. We assume, that $\theta_v$, $v \in \mathcal{V}_\mathcal{A}$, and $\theta_{uv} \in \mathcal{E}_\mathcal{A}$ are associated with the subgraph $\mathcal{A}$. Hence we consider the energy function $E_{\mathcal{A}, \theta}$ to be defined on $\mathcal{A}$ together with an *optimal labeling* on $\mathcal{A}$, which is the one that minimizes $E_{\mathcal{A}, \theta}$.

The following theorem formulates conditions necessary to produce an optimal labeling $x^*$ on the subgraph $\mathcal{G}$ from the optimal labelings on its mutually boundary complement subgraphs $\mathcal{A}$ and $\mathcal{B}$.

**Theorem 1.** *Let $\mathcal{A}$ be a subgraph of $\mathcal{G}$ and $\mathcal{B}$ be its boundary complement w.r.t. $\mathcal{A}$. Let $x^*_\mathcal{A}$ and $x^*_\mathcal{B}$ be labelings minimizing $E_{\mathcal{A}, \theta}$ and $E_{\mathcal{B}, \theta}$ respectively and let all nodes $v \in \mathcal{V}_\mathcal{A}$ be strictly arc consistent w.r.t. potentials $\theta$. Then from*

$$x^*_{\mathcal{A}, v} = x^*_{\mathcal{B}, v} \text{ for all } v \in \mathcal{V}_{\partial A} \tag{5}$$

*follows that the labeling $x^*$ with coordinates $x^*_v = \begin{cases} x^*_{\mathcal{A}, v}, & v \in \mathcal{A} \\ x^*_{\mathcal{B}, v}, & v \in \mathcal{B} \backslash \mathcal{A} \end{cases}$ , $v \in \mathcal{V}_\mathcal{G}$, is optimal on $\mathcal{G}$.*

*Proof.* Let $\theta$ denote potentials of the problem. Let us define other potentials $\theta'$ as $\theta'_w(x_w) := \begin{cases} 0, & w \in \mathcal{V}_{\partial A} \cup \mathcal{E}_{\partial A} \\ \theta_w(x_w), & w \notin \mathcal{V}_{\partial A} \cup \mathcal{E}_{\partial A} \end{cases}$ . Then $E_{\mathcal{G}, \theta}(x) = E_{\mathcal{A}, \theta'}(x|_\mathcal{A}) + E_{\mathcal{B}, \theta}(x|_\mathcal{B})$. From strict

**Algorithm 1**

**(1)** Solve LP and reparametrize $(\mathcal{G}, \theta) \to (\mathcal{G}, \tilde{\theta}^{\phi})$.

**(2)** Initialize: $(\mathcal{A}, \tilde{\theta}^{\phi})$ and $x^*_{\mathcal{A}, v}$ from arc consistent nodes.

**(3) repeat**
 Set $\mathcal{B}$ as a boundary complement to $\mathcal{A}$.
 Compute an optimal labeling $x^*_{\mathcal{B}}$ on $\mathcal{B}$.
 **If** $x^*_{\mathcal{A}}|_{\partial A} = x^*_{\mathcal{B}}|_{\partial A}$ **return**.
 **Else** set $\mathcal{C} := \{v \in \mathcal{V}_{\partial A} \colon x^*_{\mathcal{A}, v} \neq x^*_{\mathcal{B}, v}\}, \mathcal{A} := \mathcal{A} \backslash \mathcal{C}$

 **until** $\mathcal{C} = \emptyset$

---

arc consistency of $\theta$ over $\mathcal{A}$ directly follows that $E_{\mathcal{A}, \theta'}(x^*_{\mathcal{A}}) = \min_{x_{\mathcal{A}}} E_{\mathcal{A}, \theta'}(x_{\mathcal{A}})$. From this follows

$$\min_x E_{\mathcal{G}, \theta}(x) = \{ \min_{x_{\mathcal{A}}, x_{\mathcal{B}}} E_{\mathcal{A}, \theta'}(x_{\mathcal{A}}) + E_{\mathcal{B}, \theta}(x_{\mathcal{B}}) \quad \text{s.t. } x_{\mathcal{A}}|_{\partial A} = x_{\mathcal{B}}|_{\partial A} \}$$

$$= \min_{x'_{\partial A}} \min_{x_{\mathcal{A}} \colon x_{\mathcal{A}}|_{\partial A} = x'_{\partial A}} E_{\mathcal{A}, \theta'}(x_{\mathcal{A}}) + \min_{x_{\mathcal{B}} \colon x_{\mathcal{B}}|_{\partial A} = x'_{\partial A}} E_{\mathcal{B}, \theta}(x_{\mathcal{B}}) \geq \min_{x_{\mathcal{A}}} E_{\mathcal{A}, \theta'}(x_{\mathcal{A}}) + \min_{x_{\mathcal{B}}} E_{\mathcal{B}, \theta}(x_{\mathcal{B}})$$

$$= E_{\mathcal{A}, \theta'}(x^*_{\mathcal{A}}) + E_{\mathcal{B}, \theta}(x^*_{\mathcal{B}}) = E_{\mathcal{G}, \theta}(x^*)$$

$\square$

Now we are ready to transform the idea described in the introduction into Algorithm 1.

**Step (1).** As a first step of the algorithm we run an LP solver for the dual problem (3) on the whole graph $\mathcal{G}$. The output of the algorithm is the reparametrization $\tilde{\theta}^{\phi}$ of the initial problem. Since well-scalable algorithms for the dual problem (3) attain the optimum only in the limit after a potentially infinite number of iterations, we cannot afford to solve it exactly. Fortunately, it is not needed to do so and it is enough to get only a sufficiently good approximation. We will return to this point at the end of this section.

**Step (2).** We assign to the set $\mathcal{V}_{\mathcal{A}}$ the nodes of the graph $\mathcal{G}$, which satisfy the strict arc consistency condition. The optimal labeling on $\mathcal{A}$ can be trivially computed from the reparametrized unary potentials $\tilde{\theta}^{\phi}_v$ by $x^*_{\mathcal{A}, v} := \arg\min_{x_v} \tilde{\theta}^{\phi}_v(x_v), \ v \in \mathcal{A}$.

**Step (3).** We define $\mathcal{B}$ as the boundary complement to $\mathcal{A}$ w.r.t. the master graph $\mathcal{G}$ and find an optimal labeling $x^*_{\mathcal{B}}$ on the subgraph $\mathcal{B}$ with a combinatorial solver. If the boundary condition (5) holds we have found the optimal labeling according to Theorem 1. Otherwise we remove the nodes where this condition fails from $\mathcal{A}$ and repeat the whole step until either (5) holds or $\mathcal{B} = \mathcal{G}$.

### 3.1 Remarks on Algorithm 1

**Encouraging boundary consistency condition.** It is quite unlikely, that the optimal boundary labeling $x^*_{\mathcal{A}}|_{\partial A}$ obtained based only on the subgraph $\mathcal{A}$ coincides with the boundary labeling $x^*_{\mathcal{B}}|_{\partial A}$ obtained for the subgraph $\mathcal{B}$. To satisfy this condition the unary potentials should be quite strong on the border. In other words, they should be at least strictly arc consistent. Indeed they are so, since we consider the reparametrized potentials $\tilde{\theta}^{\phi}$, obtained at the LP presolve step of the algorithm.

**Single run of LP solver.** Reparametrization allows also to perform only a single run of the LP solver, keeping the results as if the subproblem over $\mathcal{A}$ has been solved at each iteration. The following theorem states this property formally.

**Theorem 2.** *Let all nodes of a graph $\mathcal{A}$ be strictly arc consistent w.r.t. potentials $\tilde{\theta}^{\phi}$, $x$ be the optimum of $E_{\mathcal{A}, \tilde{\theta}^{\phi}}$ and $\mathcal{A}'$ be a subgraph of $\mathcal{A}$. Then $x|_{\mathcal{A}'}$ optimizes $E_{\mathcal{A}', \tilde{\theta}^{\phi}}$.*

*Proof.* The proof follows directly from Definition 1. Equation (4) holds for the labeling $x|_{\mathcal{A}'}$ plugged in place of $x'$ and graph $\mathcal{A}'$ in place of $\mathcal{G}$. Hence $x|_{\mathcal{A}'}$ provides a minimum of $E_{\mathcal{A}', \tilde{\theta}^{\phi}}$. $\square$

**Presolving $\mathcal{B}$ for combinatorial solver.** Many combinatorial solvers use linear programming relaxations as a presolving step. Reparametrization of the subproblem over the subgraph $\mathcal{B}$ plays the role of such a presolver, since the optimal reparametrization corresponds to the solution of the dual problem and makes solving the primal one easier.

**Connected components analysis.** It is often the case that the subgraph $\mathcal{B}$ consists of several connected components. We apply the combinatorial solver to each of them independently.

| Dataset | | | Step (1) LP (TRWS) | | | Step (3) ILP (CPLEX) | | | $|\mathcal{B}|$ | |
|---|---|---|---|---|---|---|---|---|---|---|
| name | $|\mathcal{V}_\mathcal{G}|$ | $|\mathcal{X}_v|$ | # it | time, s | $E$ | # it | time, s | $E$ | min | max |
| tsukuba | 110592 | 16 | 250 | 186 | 369537 | 24 | 36 | 369218 | 130 | 656 |
| venus | 166222 | 20 | 2000 | 3083 | 3048296 | 10 | 69 | 3048043 | 66 | 233 |
| teddy | 168750 | 60 | 10000 | 14763 | 1345214 | 1 | – | – | 2062 | – |
| family | 425632 | 5 | 10000 | 20156 | 184825 | 18 | 2 | 184813 | 11 | 109 |
| pano | 514080 | 7 | 10000 | 34092 | 169224 | 1 | – | – | 24474 | – |

Table 1: Results on Middlebury datasets. The column **Dataset** contains the dataset name, numbers $|\mathcal{V}_\mathcal{G}|$ of nodes and $|\mathcal{X}_v|$ of labels. Columns **Step (1)** and **Step (3)** contain number of iterations, time and attained energy at steps (1) and (3) of Algorithm 1, corresponding to solving the LP relaxation and use of a combinatorial solver respectively. The column $|\mathcal{B}|$ presents starting and final sizes of the "combinatorial" subgraph $\mathcal{B}$. Dash "-" stands for failure of CPLEX, due to the size of the combinatorial subproblem.

**Subgraph $\mathcal{B}$ growing strategy.** One can consider different strategies for increasing the subgraph $\mathcal{B}$, if the boundary condition (5) does not hold. Our greedy strategy is just one possible option.
**Optimality of reparametrization.** As one can see, the reparametrization plays a significant role for our algorithm: it (i) is required for Theorem 1 to hold; (ii) serves as a criterion for the initial splitting of $\mathcal{G}$ into $\mathcal{A}$ and $\mathcal{B}$; (iii) makes the local potentials on the border $\partial A$ stronger; (iv) allows to avoid multiple runs of the LP solver, when the subgraph $\mathcal{A}$ shrinks; (v) can speed-up some combinatorial solvers by serving as a presolve result. However, there is no real reason to search for an optimal reparametrization: all its mentioned functionality remains valid also if it is non-optimal. Of course, one pays a certain price for the non-optimality: (i) the initial subgraph $\mathcal{B}$ becomes larger; (ii) the local potentials – weaker; (iii) the presolve results for the combinatorial solver become less precise. Note that even for non-optimal reparametrizations Theorem 2 holds and we need to run the LP solver only once.

## 4 Experimental evaluation

We tested our approach on problems from the Middlebury energy minimization benchmark [26] and the recently published discrete energy minimization benchmark [15], which includes the datasets from the first one. We have selected computer vision benchmarks intentionally, because many problems in this area fulfill our requirements: the underlying graph is sparse (typically it has a grid structure) and the LP relaxation delivers good practical results.

Since our experiments serve mainly as proof of concept we used general, though not always the most efficient solvers: TRW-S [16] as the LP-solver and CPLEX [27] as the combinatorial one within the OpenGM framework [28]. Unfortunately the original version of TRW-S does not provide information about strict arc consistency and does not output a reparametrization. Therefore we used our own implementation in the experiments. Depending on the type of the pairwise factors (Potts, truncated $\ell_2$ or $\ell_1$-norm) we found our implementation up to an order of magnitude slower than the freely available code of V. Kolmogorov. This fact suggests that the provided processing time can be significantly improved in more efficient future implementations.

In the first round of our experiments we considered problems (i.e. graphical models with the specified unary and pairwise factors) of the Middlebury MRF benchmark, most of which remained unsolved, to the best of our knowledge.

**MRF stereo** dataset consists of 3 models: tsukuba, venus and teddy. Since the optimal integral solution of tsukuba was recently obtained by LP-solvers [11,13], we used this dataset to show how our approach performs for clearly non-optimal reparametrizations. For this we run TRW-S for 250 iterations only. The size of the subgraph $\mathcal{B}$ grew from 130 to 656 nodes out of more than 100000 nodes of the original problem (see Table 1). On venus we obtained an optimal labeling after 10 iterations of our algorithm. During these iterations the size of the set $\mathcal{B}$ grew from 66 to 233 nodes, which is only 0.14% of the original problem size. The dataset teddy remains unsolved: though

| Dataset | $E_{\mathcal{G},\theta}(x^*)$ | Step (1) LP | | Step (3) ILP | | MCA | MPLP | | |
|---|---|---|---|---|---|---|---|---|---|
| | | # it | time, s | # it | time, s | time, s | # LP it | LP time, s | ILP time, s |
| pfau | 24010.44 | 1000 | 276 | 14 | 14 | > 55496 | 10000 | > 15000 | |
| palm | 12253.75 | 200 | 65 | 17 | 93 | 561 | 700 | 1579 | 3701 |
| clownfish | 14794.18 | 100 | 32 | 8 | 10 | 328 | 350 | 790 | 181 |
| crops | 11853.12 | 100 | 32 | 6 | 6 | 355 | 350 | 797 | 1601 |
| strawberry | 11766.34 | 100 | 29 | 8 | 31 | 483 | 350 | 697 | 1114 |

Table 2: Exemplary Potts model comparison. Datasets taken from the **Color segmentation (N8)** set. Column $E_{\mathcal{G},\theta}(x^*)$ shows the optimal energy value, columns **Step (1) LP** and **Step (3) ILP** contain number of iterations and time spent at the steps (1) and (3) of Algorithm 1, corresponding to solving the LP relaxation and use of a combinatorial solver respectively. The column **MCA** stands for the time of the multiway-cut solver reported in [21]. The **MPLP** [17] column provides number of iterations and time of the LP presolve and the time of the tightening cutting plane phase (ILP).

the size of the problem was reduced from the original 168750 to 2062 nodes, they constituted a non-manageable task for CPLEX, presumably because of the big number of labels, 60 in each node.

**MRF photomontage** models are difficult for dual solvers like TRW-S because their range of values in pairwise factors is quite large and varies from 0 to more than 500000 in a factor. Hence we used 10000 iterations of TRW-S at the first step of Algorithm 1. For the `family` dataset the algorithm decreased the size of the problem for CPLEX from originally over 400000 nodes to slightly more than 100 and found a solution of the whole problem. In contrast to `family` the initial subgraph $\mathcal{B}$ for the `panorama` dataset is much larger (about 25000 nodes) and CPLEX gave up.

**MRF inpainting.** Though applying TRW-S to both datasets `penguin` and `house` allows to decrease the problem to about $0.5\%$ of its original size, the resulting subgraphs $\mathcal{B}$ of respectively 141 and 856 nodes were too large for CPLEX, presumably because of the big number (256) of labels.

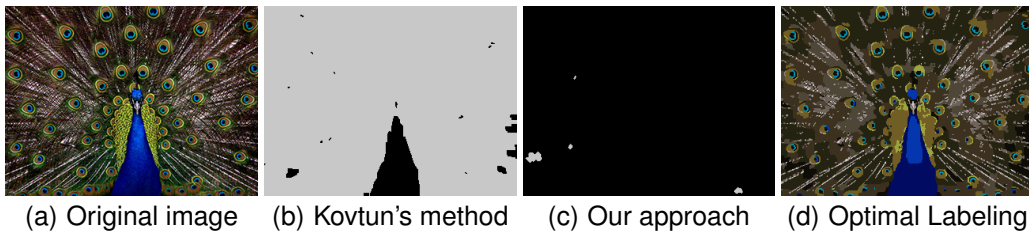

| (a) Original image | (b) Kovtun's method | (c) Our approach | (d) Optimal Labeling |

Figure 2: Results for the `pfau`-instance from [15]. Gray pixels in **(b)** and **(c)** mark nodes that need to be labeled by the combinatorial solver. Our approach **(c)** leads to much smaller combinatorial problem instances than Kovtun's method [29] **(b)** used in [30]. While Kovtun's method gets partial optimality for 5% of the nodes only, our approach requires to solve only tiny problems by a combinatorial solver.

**Potts models.** Our approach appeared to be especially efficient for Potts models. We tested it on the following datasets from the benchmark [15]: **Color segmentation (N4)**, **Color segmentation (N8)**, **Color segmentation**, **Brain** and managed to solve *all* 26 problem instances to optimality. Solving Potts models to optimality is not a big issue anymore due to the recent work [21], which related this problems to the multiway-cut problem [31] and adopted a quite efficient solver based on the cutting plane technique. However, we were able to *outperform* even this specialized solver on hard instances, which we collected in Table 2. There is indeed a simple explanation for this phenomenon: the difficult instances are those, for which the optimal labeling contains many small areas corresponding to different labels, see e.g. Fig. 2. This is not very typical for Potts models, where an optimal labeling typically consists of a small number of large segments. Since the number of cutting planes, which have to be processed by the multiway-cut solver, grows with the total length of the segment borders, the overall performance significantly drops on such instances. Our approach is able to correctly label most of the borders when solving the LP relaxation. Since the resulting subgraph $\mathcal{B}$, passed to the combinatorial solver, is quite small, the corresponding subproblems appear

easy to solve even for a general-purpose solver like CPLEX. Indeed, we expect an increase in the overall performance of our method if the multiway-cut solver would be used in place of CPLEX.

For Potts models there exist methods [29,32] providing part of an optimal solution, known as partial optimality. Often they allow to drastically simplify the problem so that it can be solved to global optimality on the remaining variables very fast, see [30]. However for hard instances like **pfau** these methods can label only a small fraction of graph nodes persistently, hence combinatorial solvers cannot solve the rest, or require a lot of time. Our method does not provide partially optimal variables: if it cannot solve the whole problem no node can be labelled as optimal at all. On the upside the subgraph $\mathcal{B}$ which is given to a combinatorial solver is typically much smaller, see Fig. 2.

For **comparison we tested the MPLP** solver [17], which is based on coordinate descent LP iterations and tightens the LP relaxation with the cutting plane approach described in [33]. We used its publicly available code [34]. However this solver did *not* managed to solve any of the considered difficult problems (marked as unsolved in the OpenGM Benchmark [15]), such as `color-seg-n8/pfau`, `mrf_stereo/{venus, teddy}`, `mrf_photomontage/{family, pano}`. For easier instances of the Potts model, we found our solver an order of magnitude faster than MPLP (see Table 2 for the exemplary comparison), though we tried different numbers of LP presolve iterations to speed up the MPLP.

**Summary.** Our experiments show that our method used even with quite general and not always the most efficient solvers like TRW-S and CPLEX allows to (i) find globally optimal solutions of large scale problem instances, which were previously unsolvable; (ii) solve hard instances of Potts models an order of magnitude faster than with a modern specialized combinatorial multiway-cut method; (iii) overcome the cutting-plane based MPLP method on the tested datasets.

## 5    Conclusions and future work

The method proposed in this paper provides a novel way of combining convex and combinatorial algorithms to solve large scale optimization problems to a global optimum. It does an efficient extraction of the subgraph, where the LP relaxation is not tight and combinatorial algorithms have to be applied. Since this subgraph often corresponds to only a tiny fraction of the initial problem, the combinatorial search becomes feasible. The method is very generic: any linear programming and combinatorial solvers can be used to carry out the respective steps of Algorithm 1. It is particularly efficient for sparse graphs and when the LP relaxation is *almost* tight.

In the future we plan to generalize the method to higher order models, tighter convex relaxations for the convex part of our solver and apply alternative and specialized solvers both for the convex and the combinatorial parts of our approach.

**Acknowledgement.** This work has been supported by the German Research Foundation (DFG) within the program Spatio-/Temporal Graphical Models and Applications in Image Analysis, grant GRK 1653. Authors thank A. Shekhovtsov, B. Flach, T. Werner, K. Antoniuk and V. Franc from the Center for Machine Perception of the Czech Technical University in Prague for fruitful discussions.

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
