[Reviews · NeurIPS 2013]

Submitted by Assigned_Reviewer_5

This paper proposes an approach to exact energy minimization in discrete graphical models.
The key idea is as follows: The LP relaxation of the problem allows to identify, via arc consistency/weak tree agreement, nodes for which the LP solution is also optimal in the discrete sense. The nodes for which discrete optimality cannot be established from the solution of the LP then define a subproblem, a hopefully small graph, which is solved exactly using a combinatorial solver. One of the contributions of the paper is to show that, if the combinatorial solution's boundary is consistent with the optimal part of the LP solution, the global optimum has been established. If the condition is not met, the combinatorial search area must be grown by the set of variables for which boundary consistency does not hold.

The proposed approach is viable only for sparse graphs, because in this special case the combinatorial search area will often be small.

Quality:
The paper is technically sound. The claims are supported by theory, and, to a lesser extent, by experiments. The current implementation of the solver is clearly not as efficient as it could be, as most of the time is spent in the supposedly very fast TRW-S algorithm. It would be interesting to see whether it would not have been faster to solve the LP using CPLEX, too, or even the whole ILP using CPLEX in some of the experiments. Such a comparison is not included in the paper (the authors argue in the rebuttal that this is not feasible).

Clarity:
The paper is clearly written and easy to understand for a reader with a background in discrete graphical models.

Originality:
The proposed approach is novel to my knowledge, although it builds heavily on theory that is well-known from the literature, such as weak tree agreement. Related work is thoroughly discussed and existing ideas are clearly attributed to the original authors.

Significance:
While the proposed approach is insightful, I do not expect it to be of significant interest to readers outside the graphical models community. Even within this community, I do not see the proposed approach becoming particularly popular, since it is not an elegant standalone approach but rather draws on "heavy machinery" (combinatorial solvers) as a black box. It provides an algorithmic perspective on solving a particular kind of problem rather than an inspiring new formulation.
Summary: This is a well-written paper that introduces a novel approach to solving discrete graphical models to optimality. The approach presented is solid, though not exciting in this reviewer's opinion, because it mostly builds on established theory.

Nonetheless, I think that the paper would be of interest to the graphical models community and hence makes for a nice contribution to the technical program of NIPS.

I'd also like to thank the authors for addressing my concerns in their rebuttal.

Submitted by Assigned_Reviewer_6

Note: I've read the reviewers' feedback. I slightly adjusted my rating based on reviewer discussions.

The paper introduces a promising new strategy for computing a globally optimal MAP solution for sparse Markov random fields. The idea is based on decomposing the overall problem in two parts: a (large) convex part and a smaller combinatorial search part. The decomposition is obtained by dividing the underlying graph of the MRF into two sub-graphs, and looking for solutions for the two sub-problems that agree on all boundary nodes that are shared. This is a very elegant idea. Similar decomposition ideas have been tried for other combinatorial optimization problems, but it is often difficult to find the right decomposition to get an overall payoff. (Another difficulty is to get the boundary assignments to "line up.") The experimental evaluation of the approach presented here shows it to be remarkably effective. Results are presented on several state-of-the-art benchmarks from computer visions. For several problems, the first globally optimal solution was found and for other instances (Potts models) the method outperforms even the best specialized techniques on the hardest instances.

The paper is very well-written. The work is well-motivated. The formal framework is well-defined and the experimental evaluation is carefully done and well-described.

Table 1 shows that the sub-graph B, capturing the combinatorics of the problem, can be surprisingly small compared to the overall graph. Can the authors provide some further intuition behind this phenomenon? I.e., what exactly is captured by B in terms of the original problem task. Is this a property of the domain or of these particular instances? Why? Also, what domain properties would break down such a nice decomposition?
Summary: A promising new technique based on a clever problem decomposition for finding globally optimal MAP solutions is presented. Convincing empirical evidence for the effectiveness of the method is provided.

Submitted by Assigned_Reviewer_7

Summary:
The paper considers MAP inference in pairwise MRFs, a problem that is generally
NP-hard. The central idea is to decompose the problem into two almost disjoint
problems, except for boundary variables, these are variables that are included
in both node sets. The central theorem of the paper says that if the two optimal
solutions given by solving both problems
independently agree on the boundary, then the overall problem is solved to
optimality. The authors introduce an algorithm that initially determines a
decomposition of the graph into two sets with a common boundary (based on arc
consistency) and then solves the larger set with LP based approaches and the
smaller problem with exact combinatorial solvers (but possibly very expensive; integer programming
solvers). If the solutions on the boundary don't agree or the LP-based approach
leads to a fractional solution, then some variables are determined with a
heuristic to be included in solving with the combinatorial method. This
procedure is reiterated till the solution agrees on the boundary.

Pros:
- Simple idea.
- Clearly written paper.
- Seems to give optimal solutions for problems that have not been solved before (did
not double-check this claim in detail).
- Nice property that the LP part of the problem does not need to be resolved,
when the combinatorial set is increased.
- Hardens the evidence that many MAP inference problems are solved almost
exactly by LP relaxations.

Cons:
- Most of the statements in the paper are relatively trivial, especially the
main theorem.
- I would have appreciated a possible comparison to cutting plane approaches
that tighten the local marginal polytope with additional constraints e.g. by
Sontag et al. UAI 2008. Software is available so it should be relatively
straightforward to compare. It might be that the solvers are too slow, in this
case I would have appreciated a remark about this.
- The paper is very short on explanations/theory on why the heuristic presented
in the paper should work, it is mostly motivated by the empirical observation
that LP solvers do a good job on many problems.
- I find the experiments lacking: the authors do not described the exact form of the
potentials they used in the experiments for the Middleburry dataset (Potts,
l1, l2, truncated ?). Does the quality of the solutions of the proposed
algorithm somehow get influenced by the form of the potentials?

Quality:
Theory is rather simple. Experiments do not match my expectations.

Clarity:
The paper is well written and easy to understand. The main idea is well
presented, I would have hoped to get a better understanding in which situations
the authors would expect the method to work well.

Originality:
Theorem 3.0.1 and 3.0.2 are relatively trivial, otherwise few original ideas
except for the combination of combinatorial and LP-based approaches.

Significance:
The paper might be significant in the sense that for a few more MAP problems the
research community knows the optimal labeling. It is however hard to judge from
the paper how practical and relevant the method is for the bulk of MAP problems.

Minor points:
067: to a convex and a combinatorial solver (no s!)
069: . in the end
060: Figure 1 is not very illustrative, what's the difference between the figure on
the left and right, why is this useful? You should explain this in the
caption.
099: ofthe
192/210: Improve numbering of remarks / theorems (just one digit)
256: Let x by a solution...
305: had to serve (why is that? use serve if you decided so)
270: would appreciate an energy comparison to the solution obtained by TRW-S
322: This might not be the final number of variables you would need to solve
for!
368: multiway-cut problem
370: for which the optimal
377: performance of our method
400: can be solved an order
Summary: Clearly written paper, relatively minor novelty and significance. Unclear for
which models the proposed method works well.
Author Feedback

Author rebuttal: We thank the reviewers for their constructive comments. R5: technically sound, claims supported by theory, clearly written and easy to understand, approach is novel and thoroughly discussed, well written paper, nice contribution. R6: Very elegant idea, experimental evaluation remarkably effective, very well written, promising new technique. R7: Clearly written paper, well written and easy to understand, might be significant.

General comments:

Reviewers question the significance of the approach.
Our approach enabled us to solve 3 instances of the widely used benchmark [24], from which only 1 instance has been solved before, as recently discussed in [14].
The fact that solvers for the subproblems can be replaced by other solvers supports existing work on dedicated solvers developed in the community, and thus flexible applicability in connection with our approach. We consider this a significant feature of our approach.

Specific comments:

* RV5: Directly applying CPLEX for the LP or solve the whole problem as an ILP.
Due to the large problem sizes this is no feasible option.
For instance, for the pfau instance, we use MCA [19] which is applicable for Potts functions and solves an equivalent ILP.

* RV6+7: Concerning the size of the subproblems in Tab. 1. / Applicability of the method / Limitations
Since the LP relaxation is quite tight and the structure sparse, we obtain small subproblems. The size of the combinatorial part B will grow for highly connected graphs or LP solutions with many fractional variable values. We do not assume that the whole LP relaxation is tight. We only exploit that the optimum of the relaxation has many integral variables. For problems in computer vision this is often the case, since local information is strong in many image regions.
We will point these issues out more clearly.

* RV7: Comparison to tighter relaxation (e.g. Sontag et. al)
We agree and will add a remark.

* RV7: Setup Middleburry dataset
We use the models provided along with [14] for reasons of reproducibility. These are identical to those used in [24].
We will point this out more clearly.

* RV7: Comparison to TRWS, results for Tab. 1.
Corresponding results are published in [14,24], e.g for venus the energy of TRWS is 3048387 and optimal is 3048043.
We will improve the presentation of this part in the final version of the paper.

* RV7: Figure 1 is not very illustrative
We will improve it.